# RoboReward: A Dataset and Benchmark for Vision-Language Reward Models in Robotics

## Abstract

A well-designed reward is critical for effective reinforcement learning-based policy improvement. In real-world robotic domains, obtaining such rewards typically requires either labor-intensive human labeling or relying on brittle hand-crafted objectives. Vision-language models (VLMs) have shown promise as automatic reward models, yet their effectiveness on real robot tasks is poorly understood. In this work, we aim to close this gap by introducing (1) **RoboReward**, a robotics reward dataset and benchmark built on large-scale real-robot corpora from Open X-Embodiment (OXE) and RoboArena, and (2) vision-language reward models trained on this dataset. Because OXE lacks failure examples, we propose counterfactual relabeling that turns successful episodes into calibrated *negative* and *near-miss* examples for the *same* video. Using this framework, we produce an extensive training and evaluation dataset, which spans diverse tasks and embodiments and enables systematic evaluation of whether state-of-the-art VLMs can provide reliable rewards for robotics. Our evaluation of the leading open-weight and proprietary VLMs reveals that no model excels in all tasks, highlighting substantial room for improvement. We then train 3B- and 7B-parameter models that outperform much larger VLMs in assigning rewards for short-horizon robotic tasks. Finally, we deploy the 3B-parameter reward VLM in real-robot reinforcement learning and find that it improves policy learning over the base 3B model by a large margin.

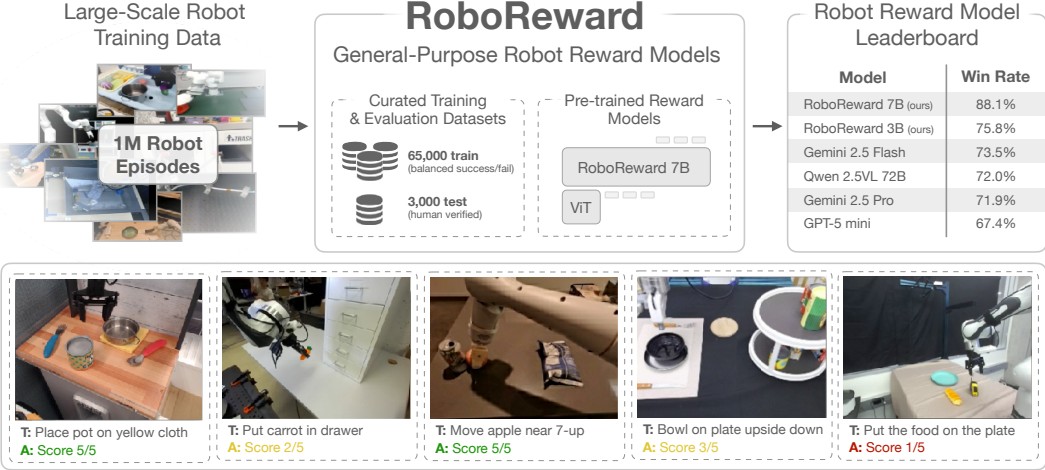

Figure 1: We introduce RoboReward, a dataset for training and evaluating general-purpose robot reward models. RoboReward consists of 3,000 real-robot episodes spanning diverse tasks and robots, with human-verified progress scores. In evaluations across 20 proprietary and open-source VLMs we demonstrate that today's models are severely lacking in their ability to provide accurate reward feedback for robots. We then curate a dataset of 65,000 scored robot episodes across diverse embodiments and train RoboReward-3B/7B, two general-purpose robot reward models that outperform all off-the-shelf models. We open-source all models, training data, and our evaluation benchmark to advance the development of general-purpose reward models for robotics.

# 1 INTRODUCTION

Despite recent algorithmic advances enabling efficient reinforcement learning (RL) training of robot control policies in the real-world (Smith et al., 2022b; Luo et al., 2024; Mark et al., 2024; Ankile et al., 2025; Chen et al., 2025b; Wagenmaker et al., 2025), the broad application of RL to real-world robotics has been severely limited by the absence of accurate and informative reward models. RL-based methods critically require a precise reward signal to direct learning, yet existing methods for obtaining such rewards typically rely on either humans to label episodes by hand (Myers et al., 2023; Wagenmaker et al., 2025), or complex and brittle hand-crafted reward functions tuned by humans through extensive trial-and-error (Lee et al., 2020; Smith et al., 2022b; Luo et al., 2024; Chen et al., 2025b). While RL as an algorithmic paradigm holds the promise of enabling automated improvement of robot policies, the need for a human in the reward design process makes modern robotic RL labor-intensive, greatly limiting its application to general, real-world robotic policy improvement.

Motivated by these challenges, recent works have explored utilizing vision-language models (VLMs) trained on internet-scale data as automated reward models for robotics (Rocamonde et al., 2023; Venuto et al., 2024; Sontakke et al., 2024; Wang et al., 2024). In principle, a highly capable VLM that can reason about the physical world could replace hand-coded heuristics and expensive human supervision. However, existing methods often fall short of achieving this, due to apparent shortcomings in current state-of-the-art VLMs and limited ability to provide sufficiently accurate rewards in real-world robot deployments. While VLMs are pretrained on large datasets drawn from a diverse set of sources—endowing them with general vision-language abilities—it is not clear that these general abilities enable them, at present, to robustly provide rewards at the level of precision and reliability required by RL training.

In this work, we seek to develop a dataset and benchmark for evaluating and improving VLM-based rewards for robotics. In simple simulation experiments, we first identify that coarse progress scores are an effective reward type for reinforcement learning, and find that reward accuracy correlates with RL performance, motivating our benchmarking design choices at a small scale before scaling up to a diverse, real robot dataset. Unfortunately, existing large-scale robotics datasets (Open X-Embodiment Collaboration et al., 2023; Khazatsky et al., 2024) are heavily skewed towards successful demonstration episodes, which are poorly suited for training and evaluating reward functions for estimating both success and failure. We therefore develop a relabeling framework for synthetically augmenting demonstration data. Our framework counterfactually relabels successful episodes with failed instructions and near-miss instructions for the *same* video, holding the video of the episode fixed while varying the commanded task. We use this technique to construct the **RoboReward** dataset, which augments the Open X-Embodiment (OXE) dataset (Open X-Embodiment Collaboration, 2023) and RoboArena evaluation benchmark dataset (Atreya et al., 2025), leading to an extensive training corpus and human-validated evaluation dataset for reward modeling across diverse tasks and embodiments (see fig. 1). Notably, our 3B and 7B vision-language reward models trained on this dataset outperform much larger VLMs, including state-of-the-art proprietary VLMs, and show promising results when used as a reward for real robot reinforcement learning.

Our contributions are as follows:

1. **Counterfactual relabeling framework.** A framework that turns success-heavy robot demonstration datasets into calibrated *negatives* and *near-misses* for the *same* videos, augmented with semantically invariant paraphrases.

2. **Robot reward benchmarking and analysis.** We first analyze supervision schemes for robotic rewards, comparing binary success signals to discrete progress labels. We also run experiments to show that higher-quality robot reward models lead to stronger downstream RL policies. We then introduce **RoboRewardBench**, a comprehensive and standardized evaluation of VLMs as reward models on full robot rollouts, where we assess 20 prominent VLMs across 3105 robot episodes spanning diverse tasks and 14 different types of embodiments.

3. **Resources.** We release the **RoboReward training dataset** and **RoboRewardBench** (evaluation dataset), **trained reward-model checkpoints** (RoboReward VLM 3B and 7B) that outperform larger VLMs on assigning rewards for short-horizon tasks, and an **evaluation suite** (including a leaderboard, prompts, raw generations, and results) to advance general-purpose reward modeling in robotics.

Our evaluation results indicate that current general-purpose VLMs are not yet reliable reward models in all settings and that the RoboReward dataset can significantly improve accuracy, taking us one step closer to fully autonomous improvement of real-world robot policies.

## 2 RELATED WORK

**Non-robot reward models.** With the recent success of RL approaches for post-training large language models (Shao et al., 2024; DeepSeek-AI et al., 2025), there has been a large number of works on training effective reward models for LLM-RL (Lightman et al., 2023; Luo et al., 2025a). Additionally, a number of benchmarks has been proposed to evaluate these language reward models. For example, RewardBench (Lambert et al., 2024) and RewardBench 2 (Malik et al., 2025) test reward model accuracy, bias, and correlation with downstream LLM-RL performance. For multimodal settings, VLRewardBench (Li et al., 2024) and Multimodal RewardBench (Yasunaga et al., 2025) probe VLM reward models across perception, hallucination, reasoning, safety, and preference judgments. In contrast to these works, our focus is on reward functions for *robotic* tasks. As our evaluations show, the capabilities of current VLMs to adequately reward robot task performance lag far behind image or text domains, motivating our RoboReward benchmark.

**Real-robot reinforcement learning.** Autonomously learning and improving robotic control policies through reinforcement learning is a longstanding goal in the robotics community. Despite limited early success applying RL directly in the real world (Riedmiller et al., 2009; Levine et al., 2016; 2018), the majority of early work in this direction focused on learning in simulation and transferring the learned policy to the real world in deployment (Cutler et al., 2014; Rajeswaran et al., 2016; Tobin et al., 2017; Peng et al., 2018; Chebotar et al., 2019; Lee et al., 2020; Kumar et al., 2021). More recently, significant progress has been made applying RL to real-world locomotion (Smith et al., 2022b;a) and manipulation (Zhu et al., 2020; Luo et al., 2024; Mendonca et al., 2024; Luo et al., 2025b) settings. These works have primarily focused on learning from scratch or with a limited number of human demonstrations, yet with the advent of "generalist" robot policies (Octo Model Team et al., 2024; Kim et al., 2024; Black et al., 2024), significant attention has been devoted to developing RL algorithms that utilize such generalist policies as a starting point for learning, improving their behavior through RL in real-world deployment (Zhang et al., 2024; Mark et al., 2024; Nakamoto et al., 2024; Chen et al., 2025b; Hu et al., 2025; Ankile et al., 2025; Wagenmaker et al., 2025; Dong et al., 2025). All of these works, however, rely on either human reward supervision or hand-crafted reward functions in order to provide a signal for learning. This has greatly limited the application of RL to general robot learning settings, a challenge we aim to resolve in this work.

**Learned reward models for robotics.** To overcome the limitations of manually specified robot rewards, there is a long line of work for *learning* robot reward functions. Early works learned robot rewards from human videos (Sermanet et al., 2016; Shao et al., 2020; Chen et al., 2021) or robot trajectories (Ma et al., 2022; 2023; Yang et al., 2023; Sontakke et al., 2024). More recent works leverage the expressivity and common-sense of VLMs to derive rewards for control. Preference-based approaches query VLMs over image and trajectory comparisons or ratings to learn reward functions and train policies in simulation or the real world (Wang et al., 2024; Venkataraman et al., 2024; Luu et al., 2025; Singh et al., 2025). A complementary direction directly derives sparse or shaped rewards from individual robot videos (Du et al., 2023; Rocamonde et al., 2023; Baumli et al., 2023; Yang et al., 2024a; Alakuijala et al., 2024; Yang et al., 2024b; Venuto et al., 2024). Ma et al. (2024) uses a VLM to perform in-context value learning. Other works target specific settings such as legged locomotion from videos (Zeng et al., 2024), text-to-video diffusion-based dense rewards (Chen et al., 2025a), autonomous driving with language-goal rewards (Huang et al., 2024), and real-to-sim iterative keypoint rewards (Patel et al., 2025). While these works demonstrate the promise of learned reward models for robotics, they typically focus on a single reward model architecture, trained for an individual robot setup. In contrast, our work presents, to our knowledge, the first comprehensive evaluation of 20 modern VLMs as *generalist* reward functions across a wide range of robot tasks and embodiments. Additionally, we provide an approach for counterfactual data relabeling that allows us to create large-scale training datasets for generalist reward functions and significantly improve over off-the-shelf models. Notably, Zhang et al. (2025) propose an alternative reward relabeling scheme based on "rewinding" robot demonstrations, but their approach disregards the content of the demonstration and is not evaluated using modern VLM models or diverse real robots.

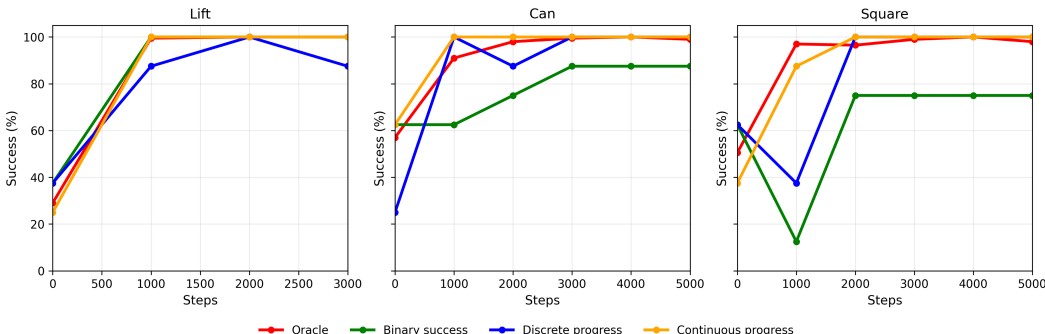

Figure 2: RL performance on three `Robomimic` tasks using learned reward functions with different reward formulations. Progress-based reward metrics lead to quicker convergence than a binary success metric. Both continuous and discrete progress rewards achieve comparably fast convergence. Thus, we choose *discrete progress* as reward type for our benchmark, since it leads to quick convergence and is easier for humans to annotate consistently than continuous progress.

Closest to our evaluation setting is the OpenGVL leaderboard (OpenGVL Team, 2025), which evaluates VLMs as temporal value estimators on expert videos via a Value–Order Correlation metric. As of September 22, 2025, OpenGVL defines two hidden tasks and reports results for ten VLMs using only successful demonstration examples. In contrast, our work evaluates 20 VLMs, measuring their ability to predict rewards (rather than values) on a range of successful *and unsuccessful* trajectories, across diverse tasks and embodiments. We also release the prompts with videos and raw model predictions alongside our leaderboard for full transparency.

## 3  THE ROLE OF REWARD IN REINFORCEMENT LEARNING

Reinforcement learning aims to find a policy $\pi$—a mapping from states to actions—that maximizes some reward $r$, typically a function of state and action. Formally, we want to find a policy $\pi$ with maximum *expected* reward: $V^\pi := \mathbb{E}^\pi[\sum_{t \geq 0} \gamma^t r_t]$, where $\gamma \in [0, 1)$ denotes a discount factor, and $r_t$ is the reward at step $t$. In practice, reward functions must be specified such that the policy learned by RL—the policy maximizing $V^\pi$—correctly achieves the desired objective.

Our goal is to design a dataset for training and evaluating learned *generalist* reward functions in robotics. The first step is to choose a reward function *type* for our evaluation. For the purpose of this work, we restrict our investigation to *episodic* rewards, which assign a reward value to a full episode rather than each individual step, and have become the standard choice of reward in many applications of RL to robotics (Luo et al., 2024; Mark et al., 2024; Ankile et al., 2025; Chen et al., 2025b; Wagenmaker et al., 2025). Still, many design choices remain: episodic rewards can be binary or multi-valued, discrete or continuous. To guide the design of our **RoboReward** benchmark, we first investigate how the choice of the reward formulation affects downstream RL performance in simulated RL tasks. Concretely, we use the `Robomimic` benchmark (Mandlekar et al., 2021), a simulation suite that includes several robotic manipulation tasks simulating common real-world robotic tasks. We seek to understand (a) what *type* of reward leads to RL training that quickly learns new tasks and (b) what is the correlation between the *accuracy* of a learned reward model and the online RL performance. In all experiments, we utilize DSRL (Wagenmaker et al., 2025)—a state-of-the-art RL fine-tuning algorithm—as our RL algorithm and apply it to finetune a diffusion policy pretrained on a dataset of task demonstrations included in `Robomimic` and ground truth rewards given by the simulation environment.

**Which reward type leads to fast RL convergence?**  We first explore what type of reward leads to the most effective RL performance. In particular, as we are primarily interested in automated, learned reward models in this work, we seek to understand what type of *learned* reward leads to the most effective RL performance. We consider three different reward types:

1. **Binary success**: Reward is 1 if the robot episode succeeds, and 0 otherwise.

2. **Continuous progress**: Reward is a continuous value in [0,1] corresponding to task progress given by the simulation environment.

3. **Discrete progress**: Similar to the continuous progress reward, but we discretize progress scores into 5 bins, and provide a reward in $\{1, \ldots, 5\}$.

For each reward type, we annotate the simulated `Robomimic` datasets with ground truth reward labels assigned by the simulation environment programmatically and finetune a Qwen2.5-VL model (Bai et al., 2025b) to predict the reward given the video of an entire episode [1].

The RL finetuning results are given in fig. 2, where we plot the true success rate against the number of samples taken. We also plot the success rate of a policy finetuned with ground-truth (binary) rewards. We see that the type of reward has significant impact on RL performance. In particular, while both learned progress rewards perform nearly as well as the ground truth rewards, the learned binary reward performs significantly worse. This suggests that learning a progress reward for effective downstream RL performance is easier than learning a success reward and, furthermore, that whether this progress reward is discrete or continuous has minimal effect on RL performance. Thus, we choose *discrete progress* as the reward formulation for RoboReward—we aim to learn a reward model that provides a progress score for a given task in $\{1, \ldots, 5\}$—since it is easier for humans to annotate consistently than fully continuous rewards.

**Do more accurate reward models lead to higher downstream RL performance?** Next, we consider how the *accuracy* of the learned reward model affects RL performance. We quantify accuracy with *mean absolute error* (MAE), the average L1 distance between predicted and ground truth rewards. Focusing on the discrete progress score reward from above, we measure reward accuracy on a held-out set of `Robomimic` validation episodes for multiple reward model checkpoints at different stages of convergence, as well as the off-the-shelf base model checkpoint. We then run RL to convergence with these reward models across all three `Robomimic` tasks. We show policy performance as a function of reward accuracy in fig. 3, where the x-axis plots the maximum possible MAE minus the model's MAE (larger values mean higher accuracy). There is a clear correlation ($r = 0.83$): more accurate rewards lead to better RL performance across the board. **These results suggest that evaluating the accuracy of a reward model on a held-out offline dataset is an effective signal for determining the performance of a downstream RL application that utilizes this reward model.**

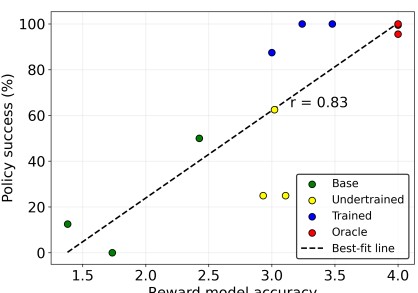

Figure 3: There is a strong positive correlation between the accuracy of learned reward models and downstream RL performance using these rewards. This validates our offline reward benchmark.

## 4 THE ROBOREWARD DATASET AND BENCHMARK

In order to train and evaluate highly capable general reward models for robotics, we need a diverse dataset of real-world robot episodes that span successful and failed rollouts and cover a wide range of tasks and embodiments. In recent years, multiple diverse real-robot datasets have been open-sourced (Open X-Embodiment Collaboration et al., 2023; Khazatsky et al., 2024; Walke et al., 2023; Fang et al., 2024; Mandlekar et al., 2018; Jiang et al., 2024; Bharadhwaj et al., 2024; Bu et al., 2025). However, most of these datasets are dominated by *successful* demonstrations collected with expert policies or humans. Although this is useful for training policies with behavioral cloning, it is suboptimal for training *reward models* that must discriminate fine-grained partial progress and failure. To address this imbalance, we introduce a counterfactual relabeling framework that can convert robot demonstration episodes into synthetic episodes with *partial success* or *failure*, thereby broadening the coverage of our reward model training corpus. Our approach is loosely inspired by the popular hindsight experience relabeling technique (HER, Andrychowicz et al. (2017)), but instead of relabeling failed episodes as successes to increase the number of successful trials, we perform "inverse-HER" and relabel successes as failures to increase the number of unsuccessful trials and

---

[1] `Robomimic` environments are Markovian, so the final state is sufficient to determine the reward.

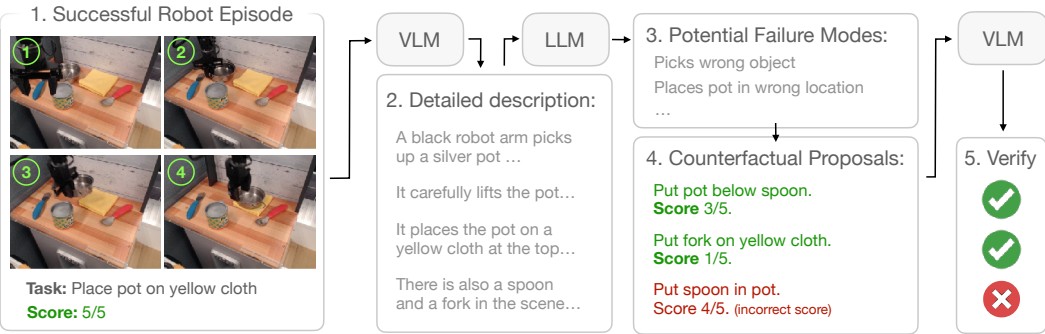

Figure 4: Overview of our counterfactual relabeling approach for generating *partial success* and *failure* task-video pairs for reward model training and evaluation. Given a successful robot episode, we use a VLM to describe it in detail, and then a sequence of LLM calls to propose alternative instructions for which the same video would result in only partial success or failure scores. A final VLM call verifies the quality of generated labels and rejects invalid labels (e.g., because the score doesn't reflect what happened in the video).

balance our training dataset. In this section we describe the data sources we use to curate our RoboReward dataset, then detail our relabeling procedure, and finally discuss the reward benchmark and models we train based on the RoboReward dataset.

## 4.1 DATA SOURCES

We aggregate real-robot videos from two primary data sources: the Open X-Embodiment dataset (OXE, Open X-Embodiment Collaboration et al. (2023)) and RoboArena (Atreya et al., 2025) evaluation data. Open X-Embodiment consists of approximately 1M real robot demonstrations, spanning 22 robot embodiments and numerous tasks, aggregated from a large number of individual academic and industry robot datasets. Since many of the datasets in OXE are highly repetitive (most demonstrations for an individual dataset may be collected in a single scene and task setup), we subsample a total maximum of 1350 episodes from each the dataset uniformly to reduce overfitting. Since all OXE episodes in our dataset are demonstrations, we assign them with the maximum reward score of 5.

RoboArena on the other hand, is a diverse dataset of real-world robot policy evaluations across a broad range of scenes and tasks, using the DROID robot platform (Khazatsky et al., 2024). Since there is comparably less repetition in RoboArena, and the dataset consists of a healthy mix of successful and failed policy rollouts, we opt to use the full dataset without subsampling. For each episode, we leverage the provided human progress score (originally in range $[0, 100]$) and map it to discrete $1 \ldots 5$ rewards. For a complete list of all RoboReward data sources and their quantities, see table 2.

## 4.2 DATA CLEANING AND COUNTERFACTUAL RELABELING

We now describe the different components of our data cleaning and counterfactual relabeling framework.

**Prompt Rewriting.** First, we normalize spelling and grammar without altering semantics, e.g., fixing spelling mistakes such as *"palce dishes in the dish rack"*. We apply a text-only rewrite transform that enforces semantic invariance: it preserves the meaning while improving the surface form. We use Qwen3 Instruct (4B) (Team, 2025) for this transform (for the exact prompt, see section B).

**Negative Example Generation.** Next, we address the imbalance of success vs failure episodes in the data. Concretely, we propose a relabeling approach that, given a successful robot rollout video, generates *counterfactual* task commands for which the same video only achieves partial success, or no success at all (see fig. 4). For example, given a video of a robot placing a pepper in a pot

on the stove top, our pipeline may generate alternatives commands `place pepper in the shelf` (partial success, since pepper was picked up), or `clean the pot on the stove` (no success). This way, we can obtain a much richer reward training dataset with a balanced distribution of successful and failed instruction-video pairs, and encourage reward models to pay close attention to the task instruction.

More formally, given an episode $e = (v, t, r)$ consisting of robot video $v$, task text $t$, and reward $r = 5$ (expert success), our pipeline constructs a calibrated set of additional training triples with modified task strings $\tilde{t}$ and labels $\tilde{r} \in \{1, 2, 3, 4\}$ for each example. Specifically, we synthesize four task commands $\{\tilde{t}^{(k)}\}_{k=1}^4$ that are grounded in visible objects and relations and calibrated so that the *same* video would plausibly score $k$ under the following end-state rubric:

- *No success* (1): The final state shows no goal-relevant change for the task command.
- *Minimal progress* (2): The final state shows a small but insufficient change toward the goal.
- *Partial completion* (3): The final state is in the general goal region but violates requirements that make it not a success.
- *Near completion* (4): The final state is correct in region and intent but misses a precise tolerance or requirement.
- *Perfect completion* (5): The final state satisfies all the requirements.

The procedure to generate the counterfactual instructions is multi-stage:

1. **Video analysis**: We use a video language model (Qwen2.5-VL Instruct 7B, Bai et al. (2025b)) to summarize the scene, the set of objects seen throughout the video and their final states.
2. **Planning**: With the video analysis, an LLM (Qwen3 Instruct 4B) proposes distinct, concrete failure modes that produce a strict ordering $1 < 2 < 3 < 4 < 5$.
3. **Command generation**: Next, the LLM proposes one imperative command per score.
4. **Verification**: The VLM checks the proposed *set* $\{\tilde{t}^{(1...4)}, t\}$ against the video of the episode and end-state rubric, returning a single *yes* or *no* verdict. Failure triggers regeneration of a set of tasks.

This relabeling procedure converts success videos into a balanced ladder of outcomes without fabricating videos. It allows us to expand our training corpus 5-fold, and our experiments demonstrate that it leads to significantly improved reward accuracy on held-out videos (section 5).

**Invariant Text Perturbation.** We further expand semantic-invariant coverage by generating multiple paraphrases $\{\hat{t}_j\}$ of each task description that preserve semantics but vary diction and syntax (e.g., synonyms) using Qwen3.

### 4.3 Training and Evaluation of General-Purpose Robot Reward Models

We split the above corpus in a training and a test set. For OXE datasets, we use the provided test set whenever defined, and otherwise split the test set off the training set. For RoboArena, we similarly split the dataset into train and test. This results in a total training set of 64850 episode-reward pairs, a validation set of 2442 and test set of 3105 samples.

We use the training set to finetune Qwen2.5 VL at two scales (3 billion and 7 billion parameters) to predict the 5-level end-of-episode progress labels when given a task description and rollout video. For both models, we freeze the vision backbone and fine-tune the fusion and LLM layers with a learning rate of $3 \times 10^{-6}$ and weight decay of $0.05$, and train with an effective batch size of 64 via gradient accumulation. For each scale, we select the best checkpoint that minimizes the mean absolute error (MAE) between the predicted and ground-truth 1-5 reward labels on a held-out validation set, producing trained vision-language reward models: **RoboReward VLM 3B** and **RoboReward VLM 7B**.

We designate the **test** split as our evaluation suite. We further refine this split by human-verifying every example — the human annotator is asked to confirm that the end-state reward label is justified

given the video of the rollout and task description. When a mismatch is found, the annotator edits the task description to reflect the reward label given the video. We refer to this verified test split as **RoboRewardBench**.

# 5 EXPERIMENTS

## 5.1 BENCHMARKING FRONTIER VLMS WITH ROBOREWARDBENCH

We evaluate 20 VLMs varying in size, model developers and access on RoboRewardBench, including our trained RoboReward VLMs. Our primary metric in *MAE* (lower is better), which is computed as the average L1 distance between the predicted reward and ground-truth label. For the overall leaderboard ranking, we order models by *mean win rate*, which is the probability that a model's score beats that of another model drawn uniformly at random in a head-to-head (see table 3).

Through this comprehensive benchmarking, we observe the following key findings:

1. **Supervision with RoboReward yields capable, compact reward models.** RoboReward VLM 7B and RoboReward VLM 3B are the top two models by mean win rate (0.881 and 0.758), followed by Gemini 2.5 Flash (0.735), Qwen 2.5 VL-Instruct 72B (0.720), and Gemini 2.5 Pro (0.719). Despite their small size, the RoboReward models beat the latest Gemini models and the largest Qwen 2.5 VL model.

2. **Generalization to unseen sources.** This pattern persists on held-out sources not in the training set. For *Austin BUDS*, the top models are RoboReward VLM 7B (MAE 0.35), Gemini 2.5 Flash (0.84), and RoboReward VLM 3B (0.99). For *NYU ROT*, RoboReward 7B and 3B are the top two (0.686 and 0.786). For *LSMO*, the top models are Gemini 2.5 Pro (0.50), RoboReward VLM 7B (0.69), and Gemini 2.0 Flash (0.78), while RoboReward VLM 3B ranks 9/20 (1.11). The only held-out dataset where RoboReward VLMs are not on top is *DLR Wheelchair Shared Control*, where GPT-5 mini leads (0.43), though RoboReward 7B/3B are close (0.60/0.63). These results indicate generalization to *unseen* scene–task pairs.

3. **Clear separation across model generations within model providers.** Gemini 2.5 Flash/Pro outperforms the previous generation of Gemini models (Gemini 2.0 Flash and Flash-Lite) with average win rates of 0.735 and 0.719 versus 0.577 and 0.491. We observe the same trend with OpenAI models: GPT-5 and GPT-5 mini outperform GPT-4.1 and GPT-4.1 mini (0.624/0.674 vs. 0.468/0.446 win rates). Within Qwen, the vision-specific VL Instruct models are stronger judges than the multimodal Omni model. This stratification demonstrates that RoboRewardBench can effectively track model progress across multiple model generations.

4. **No model is uniformly the best across all subsets of RoboReward.** The per-dataset swings in performance show that even top vision-language models underperform for certain embodiments and scenes. This echos broader findings that real-world reasoning remains challenging even for frontier VLMs (Lee et al., 2024), as reward assignment for real-world robotics is another instance of real-world reasoning.

## 5.2 TRAINING REAL-ROBOT POLICIES WITH VLM REWARD MODELS

Finally, we aim to demonstrate that RoboReward provides a sufficiently accurate reward signal to enable real-world robotic RL. For our RL algorithm, we utilize DSRL, and for our base diffusion policy which DSRL aims to improve, we train a multi-task diffusion policy on BridgeData V2 dataset (Walke et al., 2023). For a reward signal, we use a sparse end-of-episode reward, comparing the following three settings: (1) oracle human reward: a human labeler gives a positive reward of +1 on success and the reward is otherwise 0, (2) RoboReward VLM 3B: outputs a 1-5 progress score at the end of each episode, and (3) Qwen 2.5-VL Instruct 3B (base): outputs a progress score 1-5, similar to RoboReward. Both VLM rewards are prompted zero-shot.

We consider two real-world tasks on the WidowX robot. The first task is to pick up a stuffed toy mushroom and place it on a piece of cloth. The second task is to open a drawer by pulling the handle (see fig. 5). The results, obtained from 20 trials per task across the four settings, are summarized

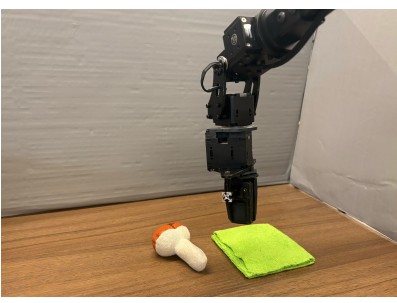 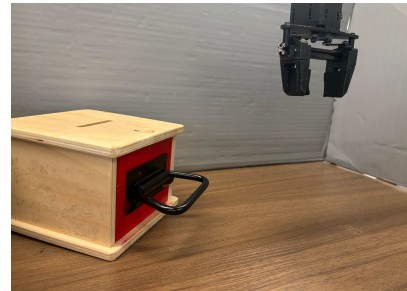

Figure 5: Real robot tasks: *Pick up the mushroom and place it on the cloth* (left) and *Grasp the black handle and pull the drawer open* (right). We use these task descriptions when prompting the VLMs to assign rewards.

Table 1: Performance of running RL with various reward models compared to the base policy (20 trials per task). Values in parentheses show the change vs. the base policy.

| Method | Pick-and-place mushroom | Open drawer |
| --- | --- | --- |
| Base diffusion policy before RL | 20% | 60% |
| DSRL + Oracle human rewards | 75% (+55) | 80% (+20) |
| DSRL + RoboReward VLM 3B (zero-shot) | 45% (+25) | 70% (+10) |
| DSRL + Qwen 2.5-VL Instruct 3B (zero-shot) | 5% (-15) | 10% (-50) |

in table 1. The base VLM reward, which acheives a lower mean win rate on RoboRewardBench (0.436), actually hurts RL performance relative to the base policy, showing that a poor reward model is worse than no RL.

On the other hand, the oracle human rewards improve performance over the base policy. In the middle is RoboReward VLM 3B (mean win rate on RoboRewardBench of 0.758), which is not human-level in assigning accurate rewards but still improves over the base policy on both tasks: pick-and-place mushroom (from $20\%$ to $45\%$ success rate over the base policy) and open drawer (from $60\%$ to $70\%$ success rate). These findings align with our results from the simulation experiments: better reward quality leads to improved downstream RL performance. This further stresses the importance of training high-quality reward models for robotics reinforcement learning. Furthermore, these results demonstrate that RoboReward is an effective reward model for enabling real-world policy improvement with RL.

## 6 DISCUSSION

In this work, we have introduced a dataset and evaluation suite, RoboRewardBench, for benchmarking generalist robot reward models, a curated dataset for training reward models, and two VLM-based reward models finetuned on this dataset which we show improve upon off-the-shelf VLMs at providing accurate rewards for robotic control settings.

While taking a first step towards providing accurate rewards for robotic tasks, this work opens the door for several interesting future directions:

- Here we have only considered short-horizon tasks, similar to those found in OXE. As robot learning continues to progress, providing rewards for longer-horizon, more complex tasks will be critical. Can we extend RoboReward to such settings?
- We have only investigated episodic rewards in this work—rewards provided at the end of an episode—but dense, step-level reward hold great promise in enabling more efficient RL, but providing a more informative learning signal through execution. What choice of dense reward is optimal, and can we utilize VLMs to obtain such dense rewards?

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

## A  THE USE OF LARGE LANGUAGE MODELS (LLMs)

We used LLMs for (i) coding assistance, (ii) copy-editing and clarity passes on text and (iii) surfacing related-work candidates that we manually vetted. LLMs were **not** used for research ideation, experimental design and analysis. All substantive research decisions and interpretations were made by the authors.

## B  DATA CLEANING AND AUGMENTATION DETAILS

### B.1  PROMPT REWRITE (INVARIANT CLEAN-UP)

**Model.** `Qwen/Qwen3-4B-Instruct-2507` (text-only).

**Purpose.** Correct grammar/spelling while preserving semantics (e.g., fix *"palce dishes in the dish rack"* to *"place the dishes in the dish rack"*).

**Prompt.**

```
Rewrite the following task description to correct grammar and spelling
↪  only.
Do not change meaning.
Task description: {TASK}
Return only the corrected text.
```

### B.2  NEGATIVE EXAMPLE GENERATION

**Models.** `Qwen/Qwen2.5-VL-7B-Instruct` (video analysis + verification) and `Qwen/Qwen3-4B-Instruct-2507` (planning + generation).

**Rubric (end-of-episode).**

```
1 - No Success: Final state shows no goal-relevant change for the
↪  command.
2 - Minimal Progress: Small but insufficient change; clearly not in a
↪  goal-appropriate region.
```

```
3 - Partial Completion: In general goal region, but a requirement makes
↪  success false
   (e.g., wrong container or orientation that breaks the goal).
4 - Near Completion: Correct region/intent but misses a precise tolerance
↪  or stability requirement
   (off-center beyond tolerance, rotated too much, not fully seated,
   ↪  unstable).
5 - Perfect Completion: All requirements within tolerances; stable after
↪  release.
```

**Video Analysis (VL).**

```
You are analyzing a video of a robot performing a short-horizon
↪  manipulation task.
Describe the scene and objects visible. Be sure to describe the objects
↪  in the task description.
Describe object positions and their relations to each other and to the
↪  robot.
Then describe, step by step, what the robot does from start to end,
↪  focusing on the final state.
Be concrete and factual. Do not invent objects that are not visible.
Task description: {ORIGINAL_TASK}
Output sections:
1) Scene and objects
2) Robot actions step by step
3) Final state summary
```

**Planning (Text).**

```
Plan carefully and step by step.
Goal: design distinct failure modes and concrete ideas for new task
↪  commands for scores 1,2,3,4
so that 1 < 2 < 3 < 4 < 5, where 5 is the original task fully satisfied
↪  by the video.
Judge only the final state and ignore time limits. Use only visible
↪  objects/relations.
Ban vague words (almost, partially, slightly, nearly, close to, near).
Each score must be strictly closer to success than the previous one.
Assign a distinct failure mode to 2, 3, and 4 (e.g., wrong region vs
↪  wrong orientation vs precision).
Original task (score 5): {ORIGINAL_TASK}
Video analysis:
{VIDEO_ANALYSIS}
Rubric:
{RUBRIC}
Produce:
1) Reasoning (what defines success for 5)
2) Separation plan (how to construct 1..4)
3) Ideas for new task commands (2-3 candidates per score)
4) Monotonicity check (why 1<2<3<4)
```

**Command Generation (Text, one score at a time).**

```
Generate a single imperative task command (one line) for the SAME video
↪  such that:
- Under the rubric it evaluates to score {K} for the final state shown.
- Stricter or different from the original; if K<5 the same video must not
↪  fully satisfy it.
- Not entailed by and not an easier subset of the original.
- Use only visible objects/relations from the analysis.
- No vague words (almost, partially, slightly, nearly, close to, near).
- Use concrete constraints (inside/on/behind, touching/not touching,
↪  left/right,
  fully inserted/contained, centered within X cm, rotation within Y
    ↪  degrees, handle orientation).
```

```
- Plain ASCII; < 25 words; start with a verb; no quotes or meta text.
Original (5): {ORIGINAL_TASK}
Video analysis:
{VIDEO_ANALYSIS}
Rubric:
{RUBRIC}
Reasoning plan:
{PLAN_TEXT}
Output only the command for score {K}, one line.
```

**Verification (VL; single decision with clear separation).**

```
Rubric:
{RUBRIC}

Set of task commands to judge for the SAME video:
Score 1: {CMD_1}
Score 2: {CMD_2}
Score 3: {CMD_3}
Score 4: {CMD_4}
Score 5 (original): {ORIGINAL_TASK}

Question:
Given the video and the rubric, do these five commands make sense and
↪   form a coherent,
strictly ordered set where the video would be graded 1,2,3,4,5
↪   respectively?

Response:
Give one brief reason.
Then write exactly one final line: ANSWER: YES or ANSWER: NO
```

## B.3 INVARIANT TEXT PERTURBATION (SEMANTICS-PRESERVING)

**Model.** Qwen/Qwen3-4B-Instruct-2507.

**Prompt.**

```
Rewrite the following task description in a different way without
↪   changing meaning.
Keep it clear. Return only the rewritten text.
Task description: {TASK}
```

## C ROBOREWARDBENCH

Table 2: Datasets used in the RoboReward dataset and benchmark.

| Name | Embodiment | Description | Train | Val | Test | Citation |
|---|---|---|---|---|---|---|
| Berkeley Bridge | WidowX | The robot interacts with household environments including kitchens, sinks, and tabletops. Skills include object rearrangement, sweeping, stacking, folding, and opening/closing doors and drawers. | 4723 | 130 | 100 | Walke et al. (2023) |
| Freiburg Franka Play | Franka | The robot interacts with toy blocks, it pick and places them, stacks them, unstacks them, opens drawers, sliding doors and turns on LED lights by pushing buttons. | 4656 | 130 | 100 | Rosete-Beas et al. (2022); Mees et al. (2023) |
| USC Jaco Play | Jaco 2 | The robot performs pick-place tasks in a tabletop toy kitchen environment. | 3898 | 130 | 260 | Dass et al. (2023) |
| Berkeley Cable Routing | Franka | The robot routes cable through a number of tight-fitting clips mounted on the table. | 4285 | 130 | 100 | Luo et al. (2023) |
| Roboturk | Sawyer | Sawyer robots flattens laundry, builds towers from bowls and searches objects. | 4940 | 130 | 100 | Mandlekar et al. (2019) |
| NYU VINN | Hello Stretch | The robot opens cabinet doors for a variety of cabinets. | 2741 | 130 | 100 | Pari et al. (2021) |
| Austin VIOLA | Franka | The robot performs various household-like tasks, such as setting up the table, or making coffee using a coffee machine. | 994 | 130 | 75 | Zhu et al. (2022a) |
| Berkeley Autolab UR5 | UR5 | The data consists of 4 robot manipulation tasks: simple pick-and-place of a stuffed animal between containers, sweeping a cloth, stacking cups, and a more difficult pick-and-place of a bottle that requires precise grasp and 6 DOF rotation. | 3587 | 130 | 100 | Chen et al. |
| TOTO Benchmark | Franka | The TOTO Benchmark Dataset contains trajectories of two tasks: scooping and pouring. For scooping, the objective is to scoop material from a bowl into the spoon. For pouring, the goal is to pour some material into a target cup on the table. | 4489 | 130 | 100 | Zhou et al. (2023) |
| NYU ROT | xArm | The robot arm performs diverse manipulation tasks on a tabletop such an box opening, cup stacking, and pouring, among others. | 0 | 0 | 70 | Haldar et al. (2023) |
| Stanford HYDRA | Franka | The robot performs the following tasks in corresponding environment: making a cup of coffee using the keurig machine; making a toast using the oven; sorting dishes onto the dish rack. | 2884 | 0 | 100 | Belkhale et al. (2023) |
| Austin BUDS | Franka | The robot is trying to solve a long-horizon kitchen task by picking up pot, placing the pot in a plate, and push them together using a picked-up tool. | 0 | 0 | 100 | Zhu et al. (2022b) |
| UCSD Kitchen | xArm | The dataset offers a comprehensive set of real-world robotic interactions, involving natural language instructions and complex manipulations with kitchen objects. | 585 | 0 | 100 | Yan et al. (2023) |
| UCSD Pick Place | xArm | The robot performs pick and place tasks in table top and kitchen scenes. The dataset contains a variety of visual variations. | 3507 | 0 | 100 | Feng et al. |
| Austin Sirius | Franka | The dataset comprises two tasks, kcup and gear. The kcup task requires opening the kcup holder, inserting the kcup into the holder, and closing the holder. The gear task requires inserting the blue gear onto the right peg, followed by inserting the smaller red gear. | 2855 | 0 | 100 | Liu et al. (2023) |
| Tokyo PR2 Fridge Opening | PR2 | PR2 opening/closing fridge and related appliance interactions. | 157 | 130 | 80 | Oh et al. (2023) |
| Tokyo PR2 Tabletop Manipulation | PR2 | Reaching, grasping, placing on PR2 across varied objects and scenes. | 1655 | 130 | 100 | Oh et al. (2023) |
| UTokyo xArm PickPlace | xArm | The robot picks up a white plate, and then places it on the red plate. | 477 | 130 | 50 | Matsushima et al. (2023) |
| UTokyo xArm Bimanual | Dual xArms | The robots reach a towel on the table. They also unfold a wrinkled towel. | 168 | 130 | 30 | Matsushima et al. (2023) |
| Berkeley MVP Data | xArm | Basic motor control tasks (reach, push, pick) on table top and toy environments (toy kitchen, toy fridge). | 2757 | 0 | 100 | Radosavovic et al. (2022) |
| Berkeley RPT Data | Franka | Picking, stacking, destacking, and bin picking with variations in objects. | 4003 | 0 | 100 | Radosavovic et al. (2023) |
| KAIST Nonprehensile Objects | Franka | The robot performs various non-prehensile manipulation tasks in a tabletop environment. It translates and reorients diverse real-world and 3d-printed objects to a target 6 dof pose. | 1258 | 0 | 100 | Salhotra et al. (2022) |
| LSMO Dataset | Cobotta | The robot avoids obstacle on the table and reaches the target object. | 0 | 0 | 210 | |
| Imperial Wrist Cam | Sawyer | CThe robot interacts with different everyday objects performing tasks such as grasping, inserting, opening, stacking, etc. | 871 | 0 | 100 | Lee et al. (2019) |
| CMU Franka Pick-Insert Data | Franka | The robot tries to pick up different shaped objects placed in front of it. It also tries to insert particular objects into a cylindrical peg. | 2980 | 0 | 100 | Saxena et al. (2023) |
| Austin Mutex | Franka | The Mutex dataset involves a diverse range of tasks in a home environment, encompassing pick and place tasks and contact-rich tasks. | 5108 | 0 | 100 | Shah et al. (2023) |
| Berkeley Fanuc Manipulation | Fanuc | A Fanuc robot performs various manipulation tasks. For example, it opens drawers, picks up objects, closes doors, closes computers, and pushes objects to desired locations. | 2549 | 0 | 100 | Radosavovic et al. (2023) |
| CMU Play Fusion | Franka | The robot plays with 3 complex scenes: a grill with many cooking objects like toaster, pan, etc. It has to pick, open, place, close. It has to set a table, move plates, cups, utensils. And it has to place dishes in the sink, dishwasher, hand cups etc. | 2921 | 0 | 100 | Lynch et al. (2023) |
| DROID | Franka | Various household manipulation tasks | 9256 | 752 | 100 | Khazatsky et al. (2024) |
| RT-1 Robot Action | Google Robot | Robot picks, places and moves 17 objects from the google micro kitchens. | 4359 | 0 | 100 | Brohan et al. (2022) |
| RoboArena | DROID (Franka-based) | Distributed real-world evaluation episodes with per-episode progress scores and pairwise preferences. | 9256 | 752 | 100 | Atreya et al. (2025) |
| DLR Wheelchair Shared Control | DLR EDAN | The robot grasps a set of different objects in a table top and a shelf. | 0 | 0 | 100 | Vogel et al. (2020) |

Table 3: Vision–language models evaluated on **RoboRewardBench** and their results. The rows are ordered by mean win rate (higher is better). *Limited* indicates restricted API-only access at the time of evaluation.

| Rank | Model | Creator | Parameters | Access | Mean win rate | Ref. |
|---|---|---|---|---|---|---|
| 1 | Qwen2.5-VL Instruct Robo Reward (7B) | This work | 7B | Open | 0.881 | – |
| 2 | Qwen2.5-VL Instruct Robo Reward (3B) | This work | 3B | Open | 0.758 | – |
| 3 | Gemini 2.5 Flash | Google | – | Limited | 0.735 | Google Cloud (2025c) |
| 4 | Qwen2.5-VL Instruct (72B) | Alibaba Group | 72B | Open | 0.720 | Bai et al. (2025a) |
| 5 | Gemini 2.5 Pro | Google | – | Limited | 0.719 | Google Cloud (2025e) |
| 6 | GPT-5 mini (2025-08-07) | OpenAI | – | Limited | 0.674 | OpenAI (2025c) |
| 7 | GPT-5 (2025-08-07) | OpenAI | – | Limited | 0.624 | OpenAI (2025b) |
| 8 | o1 (2024-12-17) | OpenAI | – | Limited | 0.590 | OpenAI (2024b) |
| 9 | Gemini 2.0 Flash | Google | – | Limited | 0.577 | Google Cloud (2025a) |
| 10 | Gemini 2.0 Flash Lite | Google | – | Limited | 0.491 | Google Cloud (2025b) |
| 11 | GPT-4.1 (2025-04-14) | OpenAI | – | Limited | 0.468 | OpenAI (2025a) |
| 12 | GPT-4.1 mini (2025-04-14) | OpenAI | – | Limited | 0.446 | OpenAI (2025a) |
| 13 | Qwen2.5-VL Instruct (3B) | Alibaba Group | 3B | Open | 0.436 | Bai et al. (2025a) |
| 14 | Qwen2.5-VL Instruct (7B) | Alibaba Group | 7B | Open | 0.378 | Bai et al. (2025a) |
| 15 | GPT-4o (2024-11-20) | OpenAI | – | Limited | 0.367 | OpenAI (2024a) |
| 16 | Qwen2.5-VL Instruct (32B) | Alibaba Group | 32B | Open | 0.321 | Bai et al. (2025a) |
| 17 | GPT-5 nano (2025-08-07) | OpenAI | – | Limited | 0.291 | OpenAI (2025c) |
| 18 | Gemini 2.5 Flash-Lite | Google | – | Limited | 0.268 | Google Cloud (2025d) |
| 19 | Qwen2.5-Omni (3B) | Alibaba Cloud | 3B | Open | 0.230 | Jin Xu (2025) |
| 20 | Qwen2.5-Omni (7B) | Alibaba Cloud | 7B | Open | 0.026 | Jin Xu (2025) |

