# OpenReview forum: "RoboReward: A Dataset and Benchmark for Vision-Language Reward Models in Robotics"
_ICLR.cc/2026/Conference — ICLR 2026 Conference Withdrawn Submission_

### Official Review · Reviewer_czRP · 2025-10-24

**Soundness:** 3
**Presentation:** 2
**Contribution:** 2
**Rating:** 4
**Confidence:** 4

**Summary:**

This paper introduces RoboReward: (1) a curated dataset and benchmark (RoboReward / RoboRewardBench) for training and evaluating vision–language reward models on short-horizon real-robot episodes, (2) a counterfactual relabeling pipeline that converts success-heavy demonstration corpora into balanced outcome ladders (1–5 progress scores) using VLM/LLM-based generation and verification, and (3) two finetuned reward VLM checkpoints (RoboReward 3B and 7B) that the authors show outperform many off-the-shelf VLMs on the benchmark and can be used as episode-level rewards to improve on-robot RL. The paper evaluates 20 VLMs on a human-verified test set (RoboRewardBench, ≈3k episodes) and reports that the trained 3B/7B models achieve top mean win rates and lower MAE; it also presents small-scale real-robot RL runs showing that RoboReward 3B improves policy success over a base diffusion policy in two WidowX tasks.

**Strengths:**

•	Assembles a large, diverse real-robot training corpus (≈64.8k pairs) and a human-verified testbed (≈3.1k examples) and releases models and leaderboards, which will be useful for benchmarking future VLM reward work.

•	The counterfactual relabeling (VLM scene summary → LLM failure-modes → command generation → verification) is a pragmatic way to create balanced outcome labels from success-only demos, enabling reward-model training without fabricating videos.

•	Benchmarks 20 VLMs across multiple held-out sources, reports MAE and a mean win-rate leaderboard, and shows compact RoboReward models can beat larger general-purpose VLMs on the task suite.

**Weaknesses:**

•	The automated counterfactual generation depends heavily on intermediate VLM/LLM judgments. The paper shows the design and a verification step, but does not quantify (in the main experiments) the fraction of generated counterfactuals accepted vs. rejected, nor the remaining label noise and its effect on model performance. Without this, it’s hard to judge how robust the dataset is to systematic mistakes in LLM proposals.

•	The benchmark and real-robot RL experiments focus on short-horizon episodic rewards; the utility for longer-horizon or dense-step rewards remains untested and is left to future work. This narrows the scope of applicability.

•	The WidowX RL runs present 20 trials per task and report point estimates (e.g., base 20% → RoboReward 45%), but do not provide variability estimates, statistical tests, or ablations (e.g., sensitivity to RL hyperparameters, reward prompting strategy, or robustness to imperfect reward labels). This weakens claims about real-world RL benefits.

•	Because a substantial fraction of training data comes from OXE (which are demonstrations often with similar setups), the degree to which the trained reward models genuinely generalize to diverse real-world deployments is mixed across held-out datasets (per-dataset MAE varies), and the authors observe that no single model is best everywhere. More analysis of failure cases and domain gaps would strengthen the claim of general-purpose reward modeling.

•	Parts of the pipeline use proprietary models (e.g., specific Qwen/Gemini/GPT APIs for paraphrasing/verification) which may limit reproducibility unless open alternatives and exact prompts are fully released. The paper indicates prompts and some settings are in the appendices, but full reproducibility depends on releasing prompt templates and code.

**Questions:**

1.	What fraction of candidate counterfactuals does the verification step accept vs. reject? Can the authors provide quantitative statistics on verification pass rates and on the human edit rate during test-set verification? This would clarify dataset noise levels.

2.	Have the authors measured how synthetic label noise (e.g., introduce x% randomly incorrect labels) affects RoboReward model MAE and downstream RL performance? This would quantify robustness to imperfect relabeling.

3.	For held-out sources where RoboReward models do poorly (or not top-ranked), can the authors provide qualitative failure examples and analyze whether failures are due to scene diversity, unseen object types, ambiguous tasks, or relabeling artifacts?

4.	Please provide exact RL hyperparameters, number of episodes used for fine-tuning, variance across runs, and whether the reward model was used zero-shot or fine-tuned for the WidowX tasks. Also, how sensitive are RL gains to prompt phrasing for the reward model?

5.	Could the authors release prompt templates, LLM/VLM model versions, and the verification script used for counterfactual generation? If parts rely on proprietary APIs, please suggest open alternatives or ablations showing how much performance depends on particular commercial models.

---

### Official Review · Reviewer_5YAv · 2025-10-26

**Soundness:** 2
**Presentation:** 3
**Contribution:** 2
**Rating:** 2
**Confidence:** 4

**Summary:**

This paper introduces a new robotics reward dataset and benchmark, “RoboReward”, constructed from large-scale real-robot corpora (from Open X‑Embodiment and RoboArena). The benchmark includes success, near-miss, and negative examples via counterfactual relabeling of successful episodes. The authors then train vision-language reward models (VLMs) on this dataset and evaluate their ability to assign rewards to robotic trajectories across diverse tasks and embodiments. They show that leading available VLMs do not excel universally; they then train 3 B- and 7 B-parameter reward models that outperform larger ones in this benchmark. Finally, they deploy a 3 B-parameter reward VLM in a real-robot RL setting and show improved policy performance over a baseline. The claim is that this benchmark and model provide a new resource for the community and empirically indicate substantial room for improvement in VLM-based reward modeling for robotics.

**Strengths:**

1. The paper tackles an important problem: reliable reward modelling for robotics is a bottleneck for applying RL in real robotic domains. RoboReward provides an approach to compare vision-language reward models in robotics.

2. The proposed way to generate more data for the existing dataset that contains successful trajectories is quite useful. The authors address the common lack of negative examples in many robotics corpora, which strengthens the dataset’s ability to train more discriminative models.

3. The evaluation demonstrates RoboReward can outperform existing VLMs, showing that fine-tuning VLMs on a specified dataset helps build the reward foundation model.

4. The paper provides ablation and analysis of model size (3 B vs 7 B), and benchmarking across tasks and embodiments, which helps future RL work build upon existing VLAs.

**Weaknesses:**

1. While the counterfactual relabelling provides a way for data augmentation that generates near-miss/failure examples, the method is too naive and not theoretically reliable. The combination of LLM and VLM may give good labels to a certain extent, but it cannot be proved that these data are 100% accurate.

2. It is unclear how realistic the assumption: "turning a successful episode into a “near-miss” or “negative” example is valid and correlates with actual failure in the environment." is in robotics and how this relabeled data compares with true failures (e.g., collisions, unstable grasps). If the assumption fails, the reward model may learn artefacts rather than true failure patterns.

3. The experiments show that the trained models outperform larger ones on short-horizon robotic tasks. However, the evaluation details are limited. For example, while the real-robot RL deployment is a plus, the details are somewhat thin. Moreover, the training details, such as GPUs and training time, are not included in the manuscript, which is not permitted.

4. It is unclear what the cost is when deploying the model to the real world. For robotics, inference time, model size, memory/compute constraints matter. If the reward model is heavy or introduces large latency, it may limit practical use.

**Questions:**

1. Why do authors fine-tune VLMs on their dataset rather than retrain a vision-language reward model? The results shown in Table 1 demonstrate that an off-the-shelf VLM (Qwen) is harmful to downstream RL training, which is even worse than the original diffusion policy. Since the VLM is only trained on the VQA dataset, does this mean that the current VLM used for question answering does not provide good prior knowledge for robotics tasks? If so, is retraining the vision language reward model on the current dataset a better option?

2. How about the training details: (for RL task) how many tasks, how many runs, what is the spread of performance across seeds?

3. Are there some examples that counterfactual relabelling provided by VLMs (reward = 0-4)? Is it realistic?

4. How about the variance (standard deviations across seeds), compute/training cost, model training hyperparameters, and failure cases when evaluating reward models in the main experiment (benchmarking)? Specifically, how many independent runs/seeds did the method use? What was the variance in success rate across runs? Can the authors provide confidence intervals and show example failure trajectories?

5. Although the authors verified on Robomimic that discrete rewards of 0-5 are comparable to continuous rewards, is this still the case in larger scenarios and more complex tasks? Why 0-5 instead of 0-10 or 0-20? Is there a trade-off between the impact of discrete numbers on RL training and the difficulty of labeling?

6. What is the inference latency and compute requirement of the trained reward model when used online in a robotics policy loop? Does this introduce any delay that could affect closed-loop control?

---

### Official Review · Reviewer_qvvQ · 2025-10-28

**Soundness:** 2
**Presentation:** 3
**Contribution:** 2
**Rating:** 2
**Confidence:** 4

**Summary:**

This paper focuses on the core bottleneck of robotic Reinforcement Learning (RL)—the challenge of acquiring high-quality reward signals. Addressing the pain points of time-consuming manual annotation and poor robustness of hand-crafted reward functions, it proposes an automated reward scheme based on Vision-Language Models (VLMs). In the method section, it first verifies the selection of reward types through simulation, then breaks down the counterfactual relabeling process. Experiments extend from offline VLM evaluation to real-world robotic RL deployment, with transparent technical details.

**Strengths:**

1. The design of reward functions is crucial in Robot Reinforcement Learning. This paper attempts to break the dilemma of binary (0-1) reward functions or manual configuration used in previous works, and proposes a paradigm of large-scale pre-trained reward models.
2. From problem formulation and method description to experimental validation, the paper features rigorous and clear logic, making it highly readable.
3. The paper commits to open-sourcing all datasets, model checkpoints, and test suites—this is of great significance to research in the entire embodied intelligence and reinforcement learning communities.

**Weaknesses:**

- ***Bias in Counterfactual Samples***: When constructing counterfactual failure cases, only the language is modified, while the original video and actions remain unchanged. The reward function trained on such data can only evaluate whether the current **successful** actions match the instructions. It fails to account for another more critical type of failure—**action failure**. Taking Figure 4 as an example, the robotic arm’s inability to grasp the pot is a more common scenario of action failure.

- ***Unreasonable Reward Evaluation***: The effectiveness of the reward function should be judged by the final success rate of the task, such as the methods demonstrated in Figure 2 and Table 1. It should not be evaluated based on its performance on the test set.

**Questions:**

1. The font size of Figures 2 and 3 is relatively small. It is recommended to adjust it to match the font size of the main text.
2. The readability of some results in Section 5.1 is extremely poor.
3. Why do Gemini and Qwen—models not trained on the dataset proposed in this paper—achieve extremely high performance (exceeding 0.7)? What might be the possible reasons?

---

### Official Review · Reviewer_cibk · 2025-10-31

**Soundness:** 3
**Presentation:** 4
**Contribution:** 2
**Rating:** 4
**Confidence:** 2

**Summary:**

VLMs, given a textual task description and video of the execution, can be used as reward models in robotics tasks. This paper introduces a dataset for comparing different VLMs as reward models. The benchmark builds on previous data (X-Embodiment, RobotArena) but extends these with negative or near-miss examples with modified task descriptions for the same video, to address the problem that most existing videos are for successful task executions. The reward labeling is done with a VLM into discrete reward categories from 1 to 5 reflecting how close the robot is from completing the given task on the video: 64850 train, 2442 validation, and 3105 test (episode,reward) pairs. The annotation for the test set of 3105 samples is further human verified. In addition, the paper fine-tunes two Qwen VLMs (3B and 7B) on these data and shows that they predict rewards more accurately for the test set and for OOD test data than off-the-shelf VLMs.

**Strengths:**

The paper is very clearly written. The need for such a benchmark dataset is justified well. The method of creating the negative and near-miss samples seems carefully designed. The models fine-tuned on the dataset outperform off-the-shelf VLMs as generic reward models. In general the quality of the work is very good.

**Weaknesses:**

I am not familiar with all recent work in reward learning in RL, hence my lower confidence score for this paper. That said, I have the some concerns/questions, and I look forward to reading the authors' replies and comments from the other reviewers.

There seems to be a trend towards dense rewards, i.e. give reward when the robot gets closer to the target during its execution, and not just the sparse (episodic) reward in the end. As the paper focuses on the latter, I'm wondering if it may be focusing a bit on the old(ish) technique?

The paper argues that the reward model accuracy and learning performance are correlated. Indeed, we are in the end interested in training a robot that can complete the task, and the reward model is only instrumental for this. Hence, why do we need to compare VLMs ability to predict the reward per se? Wouldn't it be enough to just see how well a VLM-derived reward model performs in learning?

The negatives and near-misses are done by updating the task description so that the video does not complete the updated task, while video itself stays the same. I'm wondering to what extent data generated this way reflect the data that are generated the other way round, which sounds more natural, i.e., first there's a task and and then there are videos of a robot attempting to complete the task. Is there a distribution shift between what's done in the paper and the "natural direction".

More generally, the main advancement over existing resources appears to be extend the dataset with negative and near-miss cases. While I find this useful, I wonder if this is significant enough improvement for acceptance.

**Questions:**

Small question: in Section 5.1 in the first bullet you use win-rate for comparison. In the second bullet you use MAE. Why this difference?

Small question: I did not see too many details on how the human validation for the test set was done?

---

### Note · Authors · 2025-11-18

I have read and agree with the venue's withdrawal policy on behalf of myself and my co-authors.